# Plant Phenolics: Bioavailability as a Key Determinant of Their Potential Health-Promoting Applications

**DOI:** 10.3390/antiox9121263

**Published:** 2020-12-12

**Authors:** Patricia Cosme, Ana B. Rodríguez, Javier Espino, María Garrido

**Affiliations:** Neuroimmunophysiology and Chrononutrition Research Group, Department of Physiology, Faculty of Science, University of Extremadura, 06006 Badajoz, Spain; pcosme@unex.es (P.C.); moratino@unex.es (A.B.R.)

**Keywords:** antioxidant activity, bioavailability, flavonoids, health benefits, phenolic compounds

## Abstract

Phenolic compounds are secondary metabolites widely spread throughout the plant kingdom that can be categorized as flavonoids and non-flavonoids. Interest in phenolic compounds has dramatically increased during the last decade due to their biological effects and promising therapeutic applications. In this review, we discuss the importance of phenolic compounds’ bioavailability to accomplish their physiological functions, and highlight main factors affecting such parameter throughout metabolism of phenolics, from absorption to excretion. Besides, we give an updated overview of the health benefits of phenolic compounds, which are mainly linked to both their direct (e.g., free-radical scavenging ability) and indirect (e.g., by stimulating activity of antioxidant enzymes) antioxidant properties. Such antioxidant actions reportedly help them to prevent chronic and oxidative stress-related disorders such as cancer, cardiovascular and neurodegenerative diseases, among others. Last, we comment on development of cutting-edge delivery systems intended to improve bioavailability and enhance stability of phenolic compounds in the human body.

## 1. Introduction

Phenolic compounds are secondary metabolites widely spread throughout the plant kingdom with around 8000 different phenolic structures [1]. They are involved in adaptation processes in plants during stress conditions such as wounding, infection or exposure to UV radiation [2].

In relation to their chemical structure, these compounds contain at least one phenol group [3]. This phenol is composed of an aromatic ring with one or more hydroxyl groups. Although phenolic compounds can be present in their free form in plants, they are generally present bound to sugars or proteins [4].

The interest of phenolic compounds has increased during the last decade due to their antioxidant power. Their free radical-scavenging properties help to the prevention of chronic and oxidative stress-related disorders such as cancer, cardiovascular and neurodegenerative diseases [5].

Phenolic compounds can be classified in different ways. In relation to their carbon chain, they can be divided into 16 classes (for a detailed review, see [4]). On the other hand, according to the most important classes found in the human diet, phenolic compounds are organized in phenolic acids, flavonoids and tannins. However, the classification used in this review is based on the number of phenol rings and on the structural elements linking such rings, so that phenolic compounds are categorized as flavonoids and non-flavonoids [6].

Flavonoids

Flavonoids constitute the major group of phenolic compounds. They are responsible, along with carotenoids and chlorophylls, for the blue, purple, yellow, orange and red colors in plants [7]. They have a C_6_-C_3_-C_6_ skeleton with two aromatic rings connected by a three-carbon link (Figure 1) [8]. Their antioxidant activity depends on the presence, number and position of hydroxyl groups in the chemical structure of these compounds [4]. Flavonoids can be divided into six subclasses: flavones, isoflavones, flavonols, anthocyanins, flavanols and flavanones. The differences between them are due to variations in the number and positions of the hydroxyl groups as well as in their range of alkylation and glycosylation (Figure 1) [6].

Flavones, which contain a keto group at C4, a double bond between C2 and C3, and a B ring linked to C2 (Figure 1), represent the most basic flavonoid structures. Apigenin, luteolin and their glycosides are the most abundant flavones in fruit and vegetables [9].

Isoflavones are flavones with the B ring connected to C3 (Figure 1). They are phytoestrogenic compounds found in leguminous plants. Genistein and daidzein are their principal members [1].

Flavonols are flavones hydroxylated at C3 (Figure 1) and they represent the most numerous flavonoids in fruit and vegetables. The main compounds in this subclass are myricetin, quercetin and kaempferol [9]. Cherries, grapes, apricots, red wine, chocolate and various types of tea are all rich in flavonols [5].

Anthocyanins are water-soluble pigments that provide blue, purple and red colors to vegetables. The most common are cyanidin, delphinidin, malvidin, pelargonidin, petunidin, and peonidin [8].

Flavanols comprise a complex subclass and their predominant compounds are catechin, epicatechin, gallocatechin, epigallocatechin, their 3-O-gallates, polymers and oligomers. Proanthocyanidins are oligomeric flavanols but their polymeric form are called condensed tannins [9]. Flavanols contribute to astringency, bitterness, sourness, salivary viscosity, aroma and color formation in food [1].

Flavanones compose the smallest subclass because of their limited presence, mostly, in citrus. Hesperidin is responsible of sour taste in orange juice while naringin possesses a sweeter taste [1].

Chalcones and dihydrochalcones can be also considered flavonoids because they are intermediate products in their biosynthesis. The most frequent compounds within this group are phloretin and its glucoside phloridzin, which are found in apples [4].

Non-flavonoids

Compounds with smaller and simpler chemical structures than flavonoids belong to this class. However, there are also non-flavonoids with complex structures and high molar mass [9]. Phenolic acids, coumarins, stilbenes and lignans constitute mainly this group.

Phenolic acids participate in color stability, aroma profile and antioxidant activity, which depends on the number of hydroxyl groups included in the molecule [4,5]. They are classified in hydroxybenzoic and hydroxycinnamic acids [4,5]. Hydroxybenzoic acids have a C6-C1 skeleton and are the simplest phenolic acids found in nature [1,4]. They are frequently glycosylated, joined to small organic acids or linked to structural compounds of plant cells [9]. The content of these compounds in edible plants is low, except for certain red fruits and onions [6]. Some of the most common hydroxybenzoic acids are gallic, protocatechuic, vanillic, syringic and salicylic acids [4]. On the other hand, hydroxycinnamic acids are composed by a C6-C3 skeleton and they are more common than hydroxybenzoic acids [6]. The principal dietary sources of these compounds are fruits such as apples, cherries, peaches, and citrus fruits. Some examples of hydroxybenzoic acids are coumaric, caffeic, ferulic and rosmarinic acids, *p*-coumaric and caffeic acid being the most abundant in fruits [6,8].

Coumarins are a group of phenolic compounds biosynthesized via the shikimic acid pathway, being structural derivatives of *o*-cumaric acid [10]. They are present in plants in either their free forms or as coumarin glycosides, and they absorb UV light which results in their characteristic blue fluorescence [11]. Coumarins are mostly found in olive oil, oats and spices, and their main representatives are coumarin, umbelliferone, esculetin and scopoletin [1,4].

Stilbenes have a C_6_-C_2_-C_6_ structure and are derivatives of the same biosynthetic route as flavonoids since the first part of the synthetic pathway is common to both compounds [12]. Stilbenes help to protect plant tissues from the attack of fungi, insects, and other organisms. Moreover, the synthesis of antifungal stilbenes can be induced by infections or by abiotic stimulus such as UV light [10]. Resveratrol is the most important and studied stilbene due to its antitumoral effect and can be found in grapes and wines [1].

Lignans are phytoestrogens and are synthetized by union of two cinnamic acid residues or their biogenic relatives [1]. They are present in all the organs of many vascular plant families but exhibit a low concentration in cereals, fruits, nuts, and vegetables [1,9]. An example of a plant lignan is secoisolariciresinol diglucoside. When ingested, this compound is first converted into enterodiol and eventually transformed into enterolactone by microbial enzymes in the colon [13].

## 2. Metabolism and Bioavailability of Phenolic Compounds

Phenolic compounds found in foodstuff have important health benefits and their bioavailability is a critical factor for them to accomplish their physiological function [14]. There are numerous factors affecting bioavailability. On one hand, there are factors related to the sample, such as ripening stage, cultivars or agronomic conditions, which have a direct effect on food composition. On the other hand, food processing, e.g., thermal treatments or lyophilization, and host-related factors, including age, gender, genetics, colonic microflora, and enzymatic activity, also affect bioavailability (Table 1) [15]. Likewise, bioavailability comprises different processes, including liberation from a food matrix, absorption, distribution, metabolism and elimination phases, the absorption phase being the rate-limiting factor [16].

### 2.1. The Journey of Phenolic Compounds: From Absorption to Excretion

#### 2.1.1. Small Intestine

The absorption of phenolic compounds takes place in the small intestine, specifically in the duodenum and jejunum. This process is closely linked to the cleavage and release of aglycones from the food matrix as a result of digestive enzyme activity [19]. Different enzymes are involved in this step. Thus, cytosolic β-glucosidase (CBG), which is present in the mammalian liver, kidneys and intestine, has a deglycosylation rate that is determined by the structure of phenolic aglycones and the position or nature of the sugar substitutions [20]. Another digestive enzyme is lactase-phlorizin hydrolase (LPH), which is found in the brush border portion of intestinal epithelial cells and possesses a broad specificity for flavonoid-O-β-glucosides [21]. 

Phenolic compounds can be absorbed by either passive diffusion or transporters, such as P-glycoprotein and sodium-glucose cotransporters (SGLT), present in the membrane of enterocytes. Some characteristics like molecular weight, lipophilicity, stereochemistry and presence of a group capable of hydrogen bonding can affect the transport and permeability of polyphenols (Table 1) [18]. For example, oligomeric proanthocyanidins are degraded before being absorbed due to its high molecular weight. However, gallic acid and isoflavones have low molecular weight and are readily absorbed [6,17]. Likewise, the high lipophilicity of aglycones allows them to freely cross the epithelial cells by passive diffusion, while quercetin glycosides such as quercetin-4’-O-glucoside require the action of the cotransporter SGLT-1 [22,23]. 

#### 2.1.2. Colon

Only 5–10% of total phenolic compounds are absorbed in the small intestine. Most of these compounds are transported to the colon where they are subjected to the enzymatic activity of the gut microbial community [24]. The transformation of molecules depends not only on gut microbial population, but also on chemical structure of the compound and type of food ingested. For example, daidzein, an isoflavone mainly found in soy, is transformed to equol and *O*-desmethylangolensin, but only 30–40% of soy consumers have the microflora population that produces equol [14]. Gut microbiota is composed of numerous microorganisms, has a high hydrolytic activity and its enzymes catalyze reactions of hydrolysis, dihydroxylation, demethylation and decarboxylation. Usually, polymeric phenolic compounds are degraded to phenolic acids, flavonols are decomposed into hydroxyphenylacetic acids, flavones and flavanones are broken down into hydroxy- phenylpropionic acids, and proanthocyanidins are degraded to phenolic acids of lower molecular weight [15]. Thus, thanks to the colonic microflora, non-absorbed compounds are turned into bioavailable metabolites. For instance, the absorption of pomegranate phenolics requires a previous step where gut microbiota metabolizes ellagitannins and gallotannins to produce two dibenzopyranones known as urolithin A and its monohydroxylated analogue urolithin B. These metabolites are bioavailable and exert health benefits [25]. In fact, urolithins and their conjugates can be detected in plasma at 0.5 and 0.6 h (0.04 and 0.11 μm urolithin A, respectively), with continuous excretion in urine for 48 h after pomegranate intake [26]. In addition, ferulic acid is metabolized to hydroferulic acid, isoferulic acid, vanillic acid, 3-hydroxyphenylacetic acid and protocatechuic acid by gut microbiota. Hydroferulic and 3-hydroxyphenylacetic acids can be found in rat brain along with urolithins A and B, which demonstrates the importance of these compounds as neuroprotective agents [26]. Moreover, the anxiolytic activity observed after oral administration of kaempferol and quercetin was not detected after intraperitoneal administration. This suggests that their metabolites 3,4-dihydroxyphenylacetic acid and *p*-hydroxyphenylacetic acid, produced by colonic microflora, are indeed responsible of the anxiolytic effect [27].

#### 2.1.3. Liver

From the colon, the metabolites are transported to the liver through portal vein. Once in the liver, phenolic compounds are biotransformed to make them more polar and facilitate their excretion. Biotransformation reactions are classified into phase I and phase II. Each phase includes different steps which are catalyzed by complex enzymes. Furthermore, metabolites may return to the small intestine via enterohepatic recirculation, be further absorbed by enterocytes and transported back to the liver [18,19].

Phase I, known as modification phase, comprises different structural modifications including thiolation, hydroxylation, amination, *N-/O*-dealkylation or carboxylation. Hydroxylation of hydrocarbons is catalyzed by the large and diverse family of cytochromes P450 (CYP450). In this sense, it has been observed that the presence of many hydroxyl groups on the B-ring of flavonoids such as quercetin prevents further hydroxylation [20]. In the case of resveratrol, it can be hydroxylated by CYP450 giving rise to piceatannol and genistein, while it can be mono-oxygenated by the action of CYP1A1, CYP1A2, CYP1B1 or CYP2E1 thus forming orobol [22,26].Phase II, the so-called conjugation phase, includes reactions such as glucuronidation, methylation, sulfation or a combination of them [24]. Main enzymes involved in this phase are uridine-5’-diphospho-glucurunosyltransferases (UGTs), sulfotransferases (SULTs) and catechol-*O*-methyltransferases (COMTs) [23]. The conjugations catalyzed by these enzymes can alter the bioactive properties of phenolic compounds. UGT family participates in the metabolism of resveratrol, producing resveratrol-3’-*O*-glucuronide and resveratrol-4’-*O*-glucuronide. On the other hand, resveratrol-3’-*O*-sulfate, resveratrol-4’-*O*-sulfate and resveratrol-3,4’-*O*-disulfate are the main sulfates generated by SULTs [26]. In this line, previous studies compared inhibition of cyclo-oxygenase (COX)-1 and COX-2 by resveratrol and its metabolites, bearing in mind that COX-1 and COX-2 are responsible of resveratrol’s cardioprotective and anticancer effects. The IC_50_ values revealed that both resveratrol and its metabolite resveratrol-4’-*O*-sulfate are potent inhibitors of both these enzymes, whereas resveratrol-3’-*O*-sulfate and resveratrol-3’-*O*-glucuronide are weak inhibitors. Concentrations of resveratrol and resveratrol-4’-*O*-sulfate in human plasma were reported to be 0.5 µm and 2–10 µm, respectively, after an oral dose of pure resveratrol, thereby suggesting that in vivo inhibition of COX-1 and COX-2 could be due to the sulfate derivative [28].

Once both metabolic phases have taken place, metabolites are distributed through the blood by plasma proteins, e.g., albumin, to different organs and tissues. Normally, polyphenols are present in those tissues where they has been metabolized, but they can also reach specific target tissues such as pulmonary, pancreatic, cerebral, cardiac and splenic tissues [29]. Thus, for instance, catechin and epicatechin glucuronides were found in liver and brain after 4 h (8 nmol g^−1^ and 6.4 nmol g^−1^, respectively), kidney after 2 h (2.23–5.1 nmol g^−1^), lung after 2 h (20 nmol g^−1^), muscle after 21 days (73 μmol g^−1^), and brown and white adipose tissue after 21 days of intake (157 μmol g^−1^) in rats fed with different food extracts [26]. However, derivatives of phenolic compounds can also reach the kidney to be excreted. Large and extensively conjugated metabolites are eliminated in the bile, while small conjugated metabolites, such as monosulfate derivatives, are excreted in urine [13]. Previous studies showed that urinary concentration of some catechins of tea ranged between 0.5–6% of initial intake, while concentration of epicatechin of cocoa in urine was up to 30% of initial intake [29]. Likewise, another study with a polyphenol-rich extract of exocarpium *Citrus grandis* demonstrated that excretion of naringenin metabolites was 5.45% of initial intake (250 mL) and mainly occurred within 4–12 h after its consumption [30].

### 2.2. Measuring Bioavailability: A Not-So-Easy Challenge

The biological activities of phenolic compounds may differ depending on their biotransformation routes. In this sense, an ever-growing number of studies have analyzed the bioavailability and metabolism of phenolic compounds [31]. In order to measure bioavailability, the amount of food bioactive compounds circulating in the blood stream must be determined. To do so, it is necessary to know the kinetic, i.e., the speed with which a given compound enters the blood stream from the intestine, as well as the maximum concentration reached and the time it takes to achieve such concentration. Besides, the rate with which a compound is removed from the blood due to its metabolism and its excretion should be also known [14]. For example, after consumption of green tea, flavanols are rapidly absorbed reaching their maximum concentration within 0.5 to 2 h, which is followed by a fast metabolism and excretion. Likewise, bioavailability studies of hesperidin (flavanone mostly found in citrus fruits such us oranges, lemons and grapefruits) have shown that the time when maximum concentration was reached corresponded to 4.4–7 h. However, the modification of hesperidin to hesperetin-7-glucoside increased its bioavailability and modified its absorption kinetics, thereby reaching the maximum concentration in 0.6 h [16]. Accordingly, the absorption kinetic is mainly dependent on physical and chemical characteristics of the bioactive compounds but it can be also influenced by the physiology of the subject (age, genetic profile, gender, lifestyle, etc.), thus giving rise to a unique bioavailability profile. Half-life of bioactive compounds may therefore vary from minutes (e.g., gallic acid) to hours (e.g., rutin) [14].

## 3. Biological Effects of Phenolic Compounds

Phenolic compounds have been recently widely studied due to their biological effects, which could be beneficial for human health [32]. These benefits are mainly related to both their direct and indirect antioxidant actions. In fact, polyphenols are able to donate electrons to oxidant species, scavenge free radicals and chelate metal ions [29], but can also indirectly attenuate production of reactive oxygen species (ROS) by either improving antioxidant enzymes’ activity or inhibiting enzymes that induce pro-oxidant effects [33]. For example, kaempferol depicts a high antioxidant capacity because of its tendency to donate electrons [34]. Similarly, fisetin shows protective effects against cell death, ROS scavenging actions and stimulation of glutathione antioxidant capacity, thus significantly reducing oxidative damage in lipids, DNA and proteins in Chinese hamster lung fibroblasts [35]. The same effect was observed with a grape seed extract rich in catechins, proanthocyanidins and anthocyanidins in human keratinocyte cell line HaCaT. The antioxidant power of this extract protected keratinocytes against ROS formation, thereby reducing oxidative stress, DNA damage and apoptosis, while increasing cell survival [36]. Antioxidant activity, whether direct or indirect, has also been demonstrated in other phenolic compounds such as phenolic acids and tyrosol, the former prevents metal catalysis and free radicals formation and the latter inhibits damage caused by ROS in human umbilical cord vein endothelial cells [37,38]. On the other hand, phenolic compounds also possess other biological effects related to their antioxidant capacity such as antimicrobial, anti-inflammatory, anticancer and cardioprotective activities (Figure 2). Additionally, it is important to note that phytochemical mixtures can influence the expected biological effect caused by the individual compounds. For example, it has been observed that the combination of extracts from *Potentilla fruticose* L. leaves and green tea polyphenols showed enhanced radical scavenging properties mainly due to the synergistic activities of the phenolic compound hyperoside (abundant in *P. fruticose* L. leaves) with epicatechin gallate (ECG; present in green tea) [39]. Likewise, it has been shown that the combination of chlorogenic acid and isoquercitrin improved their superoxide anion scavenging actions [40]. 

### 3.1. Antimicrobial Activity

The antimicrobial properties of phenolic compounds are due to the ability of their hydroxyl groups to bind the active sites of key enzymes and modify the metabolism of microorganisms [41]. Antimicrobial activity depends on the position of the hydroxyl substitution in the aromatic ring, as well as on the length of the saturated side-chain [42]. For example, it has been demonstrated that caffeic acid possesses higher antimicrobial activity than *p*-coumaric acid because the first one has more hydroxyl groups substituted in the phenolic ring [43]. 

Some of the investigations that examine antibacterial and antifungal activity of different phenolic compounds are highlighted in the next few lines. For instance, catechin inhibits the growth of *Helicobacter pylori* and *Escherichia coli* [44,45]. Likewise, resveratrol displays antibacterial capacity against *Enterococcus faecalis*, *Campylobacter spp.*, *Arcobacter butzleri* and *Candida albicans* [46,47,48]. Moreover, the pulp of two varieties of Portuguese red grape shows an effective growth inhibition against *Klebsiella pneumoniae*, *Staphylococcus epidermis*, *Listeria monocytogenes* and *Staphylococcus aureus* due to their content in anthocyanins and tannins [49]. Finally, essential oils from different thyme species that are rich in phenolic compounds act against microorganisms like *Pseudomonas aeruginosa*, *Cronobacter sakazakii*, *Listeria innocua*, *Streptococcus pyogenes*, *Candida albicans*, *Saccharomyces cerevisiae*, *Staphylococcus aureus* and *Salmonella enterica* [50,51].

Synergistic interactions of phenolic compounds with antibiotics have been also investigated with the aim of counteracting antibiotic resistance. Thus, licoarylcoumarin, glycycoumarin and gancaonin, which are found in licorice, were reported to evoke antibacterial effect on vancomycin-resistant strains of *Enterococcus faecium* and *Enterococcus faecalis* [52]. Furthermore, polyphenols present in Cabernet Sauvignon grape pomace were shown to potentiate the effect of different classes of antibiotics against *Staphylococcus aureus* and *Escherichia coli*, especially against multi-drug resistant clinical isolates [53].

As for in vitro antiviral effect, various studies have reported the potential of phenolic compounds. For example, it has been observed that gallic acid presented a potent effect on Herpes simplex virus type 1 (HSV-1) and parainfluenza type 3 [54]. Besides, one flavanone, naringenin, has been shown to inhibit Dengue virus replication in infected primary human monocytes [55]. Additionally, epigallocatechin gallate (EGCG), particularly abundant in green tea, was demonstrated to block Hepatitis B virus entry into immortalized human primary hepatocytes [56]. In this line, EGCG alongside delphinidin were proven to reduce the infectivity of Dengue virus, Zika virus and West Nile virus by affecting the attachment and entry steps of the viruses life cycle [57]. Both compounds have also demonstrated to impede hepatitis C virus entry in primary human hepatocytes by alteration of the viral particle structure, which hinders its attachment to the cell surface [58].

### 3.2. Anti-Inflammatory Activity

The inflammation process exacerbates ROS and reactive nitrogen species (RNS) production, thereby increasing the activity of proinflammatory agents [6]. Anti-inflammatory activity of phenolic compounds interrupts ROS-dependent inflammation cycle [59] and acts against pro-inflammatory mediators like tumor necrosis factor-α (TNF-α), interleukin (IL)-1β, -6 and -8, inducible nitric oxide synthase (iNOS), COX, and leukotrienes. Besides, phenolic compounds also regulate expression of transcription factors such as nuclear factor kappa-light-chain-enhancer of activated B cells (NK-κB) [26].

Among phenolic compounds, stilbenes are one of the most thoroughly studied compounds. In this line, various studies have shown the capacity of resveratrol to inhibit COX activity, inactivate peroxisome proliferator-activated receptor gamma (PPARγ) and induce endothelial nitric oxide synthase (eNOS) in murine and rat macrophages. Besides, a resveratrol analog, RVSA40, was proven to inhibit TNF-α and IL-6 production in RAW 264.47 (murine macrophages cell line) [59]. Likewise, a study with 25 different stilbenes revealed that piceatannol and pinostilbene demonstrated a comparable activity to the anti-inflammatory drugs zileuton and ibuprofen in inhibiting COX-1, COX-2 and 5-lipoxygenase (5-LO) activity [60]. In addition, in the same study, it was shown that the majority of those 25 compounds managed to reduce the activity of NF-κB/activator protein-1 (AP-1) and attenuate the expression of TNF-α in human monocytic leukemia cell line THP-1 [60]. Similarly, in another study, it was observed the inhibition of COX-2 by pinostilbene and of COX-1 by pinostilbene and oxyresveratrol. In relation to 5-LO, the effective inhibitors were shown to be pterolstilbene alongside oxyresveratrol [61]. 

Anti-inflammatory effects were also found in different metabolites of phenolic compounds. For example, 4’-*O*-methyl-gallic acid is a metabolite generated after fruit juice consumption that reduces the release of pro-inflammatory cytokines and inhibits the expression of COX-2 and iNOS genes by the suppression of NF-κB activation in macrophages [26]. Likewise, quercetin-3’-*O*-glucuronide proved to decrease the transcription of genes implicated in inflammation, such as pro-inflammatory interleukins and enzymes involved in oxidative stress responses [62]. Other metabolite, apigenin-7’-*O*-glucuronide, was reported to suppress the release of nitric oxide (NO) and TNF-α in lipopolysaccharide-stimulated RAW 264.47 macrophages, and also prevented lipopolysaccharide-induced mRNA expression of iNOS, COX-2 and TNF-α [63,64].

### 3.3. Anticancer Activity

Sustained oxidative stress and increased ROS levels are typical features of cancer. Phenolic compounds can interrupt or reverse carcinogenesis by both modulating intracellular signaling molecules involved in cancer initiation and/or promotion, and blocking the progression of cancer [15]. For instance, gingerol, a ginger-derived phenolic compound, and its derivative 6-shogaol show anticancer activity in brain, lung and breast cancer [65,66]. Such an anticancer ability was also demonstrated for hesperetin, which reduced cell viability and induced apoptosis in human cervical cancer SiHa cells [67].

Numerous studies have shown the important role of phenolic compounds in different types of cancer. Specifically, flavonoids were shown to hinder cell proliferation and stimulate DNA repair by reducing oxidative stress, thus preventing cancer initiation and promotion. Furthermore, in the progression stage, flavonoids were also reported to inhibit proangiogenic factors, regulate metastasis-related proteins and induce apoptosis [33]. On the other hand, EGCG was described to decrease the number and size of tumors, improve oxidative stress markers and inhibit expression of proangiogenic factor such as CD44, VEGF, Ki-67 and MMP-2 in rats bearing mammary cancer [68]. Likewise, it has been demonstrated that the combination of EGCG and curcumin produced a synergistic effect that enhanced inhibitory actions of EGCG on cell proliferation of PC3 prostate cancer cell line [69].

Phenolic compounds found in olive oil have been also widely studied in relation to the prevention of cancer risk in different tissues. For example, oleocanthal was shown to inhibit the growth of three breast cancer cell lines, namely BT-474, MCF-7 and T-47D, in mitogen-free media by decreasing nuclear expression of estrogen receptor-α. Moreover, combined treatment with oleocanthal and tamoxifen produced a synergic inhibition of cell proliferation in these cell lines, thereby suggesting that oleocanthal improves sensibility to tamoxifen treatment [70]. Likewise, hydroxytyrosol, a metabolite of oleuropein, as well as phenylacetic and hydroxyphenylpropionic acids were informed to arrest cell cycle and induce apoptosis in Caco-2 and HT-29 cell lines [71]. On the other hand, high doses of hydroxytyrosol also caused apoptosis in papillary (TPC-1 y FB-2) and follicular (WRO) thyroid cancer cell lines via mitochondrial apoptotic mechanism involving the release of cytochrome c and the up-regulation of p53 and BAD [72]. Finally, combination of oleuropein and doxorubicin produced stronger cytotoxic and apoptotic effects than individual treatment with high doses of doxorubicin in MDA-MB-231 breast cancer xenografts [73].

Regarding human trials, it has been demonstrated that consumption of freeze-dried strawberries powder for 6 months reduced the histologic grade of dysplastic premalignant lesions in 80.6% of patients with esophageal dysplastic lesions in a high-risk area for esophageal cancer. Besides, this strawberry powder also decreased expression levels of the pro-inflammatory mediator iNOS and did not cause any toxic effect [74]. In other study, the supplementation of pomegranate extract in patients with colorectal cancer had a significant down-regulating effect on the expression of colorectal cancer-related genes such as CD44, CTNNB1, CDKN1A, and EGFR in surgical colon samples [75].

### 3.4. Cardioprotective Actions

Cardiovascular diseases are mostly caused by oxidative stress and behavioral risk factors such as tobacco use, alcohol abuse, sedentary lifestyles and high-fat diets. Different studies have claimed that the consumption of phenolic-rich food reduces the risk of suffering such diseases [33]. In fact, phenolic compounds can alter lipid metabolism, prevent oxidation of low-density lipoproteins (LDL), increase levels of high-density lipoproteins (HDL) and manifest vasodilatory properties, hence decreasing the risk of coronary issues, ischemia and cardiomyopathies. These compounds also promote antiplatelet aggregation, improve endothelial function and diminish the expression of cell adhesion molecules [76]. 

Atherosclerosis is a multifactorial disease characterized by an endothelial dysfunction that involves ROS-dependent LDL oxidation (ox-LDL) [33]. In this regard, quercetin has been demonstrated to inhibit ox-LDL-induced adhesion of endothelial leukocytes by attenuating the Toll-like receptor(TLR)-NF-κB signaling pathway and decreasing the atherosclerotic inflammatory process in rats fed a hypercholesterolemic diet [77]. This flavonoid is also capable of regulating expression of NADPH oxidase subunits, which are the main source of ROS in phagocytic and vascular cells, and reducing the atherosclerotic plaque area in mouse fed a high-fat diet [78]. On the other hand, extra virgin olive oil has been proven to modulate expression of microRNAs (miRNAs), which are small non-coding RNA molecules involved in post-translational regulation of gene expression that has been found to be dysregulated in different diseases such as atherosclerosis or cancer. Specifically, consumption of extra virgin olive oil, which is rich in phenolic compounds, diminished the overexpression of miR-146b-5p (characteristic of atherosclerotic plaques), miR-769-5p and miR-192-5p (associated with the alteration of glucose metabolism, fatty liver and acute myocardial infarction) [79]. Besides, an olive extract rich in secoiridoids has shown its capacity to weaken initial steps of atherosclerosis due to significant reduction of endothelial dysfunction biomarkers such as E-selectin, VCAM-1, MCP-1, ICAM-1 and F4/80 in ApoE^−/−^ mice, an atherosclerosis-prone mouse model [80]. Moreover, epicatechin was shown to attenuate pro-atherogenic inflammatory processes in female ApoE*3-Leiden transgenic mice, a well-established model of hyperlipidemia and atherosclerosis induced by diet [81].

The cardioprotective effect of phenolic compounds has been shown in different other studies. For instance, it was observed that characteristics of metabolic syndrome induced with a high-fat, high-fructose diet in Wistar rats were attenuated by supplementation with a grape pomace that was rich in 26 different phenolic compounds [82]. Additionally, Wistar rats fed for 14 months with an extract rich in malvidin, delphinidin, rutin, quercetin, catechin, coumaric acid, kaempferol and trans-cinnamic acid prevented hypertrophy, inflammation, fibrosis and cardiomyocytes apoptosis [83]. Finally, vasorelaxant activity of a ferulic acid metabolite, ferulic acid-4-*O*-sulfate, was also demonstrated in both isolated mouse arteries and anesthetized mice [84]. 

As for human studies, two trials investigated the effect of green tea catechins on LDL oxidation. In both studies, total antioxidant capacity of plasma was increased and LDL oxidizability was reduced. These findings suggest the importance of green tea catechins intake with regards to the reduction of atherosclerosis risk [85]. Likewise, another study explored the effect of consuming phenolics-rich extra virgin olive oil in 18 healthy subjects and noticed that its consumption decreased systolic pressure, apparently, through the modulation of ACE and NR1H2 gene expression, which are related to the renin-angiotensin-aldosterone system [86].

### 3.5. Other Health Benefits

There are other beneficial effects, such as antidiabetic effects, that have been attributed to different polyphenols. Type II diabetes mellitus causes chronic oxidative stress as a consequence of hyperglycemia, insulin resistance, inflammation and dyslipidemia, and may result in defective expression of insulin gene and impaired insulin secretion [33]. Two flavanones, naringin and naringenin, have been reported to produce antidiabetic actions by enhancing expression of insulin receptor, glucose transporter GLUT4 and adiponectin in type II diabetic rats, thereby improving insulin resistance [87]. Similar results were observed with the procyanidin cinnamtannin A2, which prevented hyperglycemia and enhanced glucose tolerance by promoting GLUT4 translocation and glucose uptake [88]. In addition, other flavanone, hesperidin, has been demonstrated to improve glycemic control and reduce DNA oxidative damage and lipid peroxidation associated to hyperglycemia in type II diabetes patients [89]. 

The health benefits of phenolic compounds have been also studied in neurodegenerative diseases such as Alzheimer’s and Parkinson’s. In Alzheimer’s disease, which is characterized by accumulation of amyloid-β (Aβ) and tau aggregates, EGCG was shown to be an effective inhibitor of tau aggregation and toxicity, which could be beneficial to hinder Alzheimer’s progression [90]. Previous research in mice has also shown that quercetin administration reduced histopathological characteristics of the disease such as extracellular β-amyloidosis, tauopathy, astrogliosis and microgliosis in the hippocampus and the amygdala, while improving cognitive and emotional dysfunction [91]. Importantly, in a clinical study, it has been demonstrated that daily consumption of anthocyanin-rich cherry juice improves verbal fluency, short-term memory, and long-term memory in older adults with Alzheimer’s. Besides, both systolic and diastolic blood pressure was lower in these patients [92]. 

As for Parkinson’s, which is characterized by dopaminergic neuronal loss in substantia nigra associated to oxidative stress, neuroinflammation and apoptosis, some phenolic compounds have been proven potentially effective as therapeutic agents. Thus, the flavone apigenin was reported to decrease both α-synuclein accumulation, a protein associated with sporadic and hereditary cases of the disease, and motor deficits in a rotenone-induced rat model of Parkinson’s. Furthermore, apigenin also protected against neuronal apoptosis via regulation of tyrosine hydroxylase, and increased dopamine biosynthesis as well as the expression of dopamine D2 receptor [93]. Finally, rutin was informed to suppress the expression of genes related to dopaminergic neuronal cell death such as *Park5*, *Park7* and *Casp3*, while isoquercitrin inhibited *Park5* and *Park7* expression in rat pheochromocytoma PC12 cells [94].

### 3.6. Safety Profile of Phenolic Compounds

Although phenolic compounds reportedly possess a plethora of biological activities, it is necessary to address the issue of their safe dosage. In fact, phenolic compounds generally exhibit bimodal pharmacological effects as they can exert therapeutic actions at low doses while producing toxicity at high doses. However, current studies on the harmful effects of phenolic compounds are mostly based on cell experiments and animal models, with few studies in humans. For instance, safety studies on EGCG have established a no-observed adverse effect level (NOAEL) of 500 mg EGCG/kg/day in both rats and dogs [95]. In this line, a NOAEL of 600 mg EGCG/day and an acceptable daily intake for 70-kg adult humans of 322 mg EGCG/day were reported for humans [96]. As for resveratrol, it is quite well tolerated by experimental models, with oral doses of 200 mg/kg/day in rats and 600 mg/kg/day in dogs showing no apparent side effects [97]. Likewise, a dose of 450 mg resveratrol/day was described to be safe for a 60-kg person [98]. In the case of proanthocyanidins, toxicity tests indicated that Oligopin^®^ and Enzogenol^®^, two pine bark extracts rich in proanthocyanidins, were well tolerated following repeated oral administration to rats, with a NOAEL of 1000 mg/kg/day and 2500 mg/kg/day, respectively [99,100]. Moreover, consumption of 960 mg/day for 5 weeks in human studies suggested lack of toxicity of Enzogenol^®^ [100]. It has been also described that protocatechuic acid is well tolerated in animal models, the lethal dose 50 of protocatechuic acid being 800 mg/kg for intraperitoneal injection, 3.5 g/kg for intravenous injection, and 500 mg/kg for oral administration [101]. Therefore, it seems obvious that toxic doses of phenolic compounds are much higher than those delivered by daily food consumption; however, dosage of phenolic compounds in the context of dietary supplements may require further research in animal models and clinical trials before a definite recommendation is done.

## 4. Current Trends in Research of Phenolics Bioavailability

The incorporation of phenolic compounds into the diet is required to benefit from their biological effects. Traditionally, these compounds have been directly incorporated into foodstuff, but their instability during food processing, distribution and storage, as well as their low absorption and bioavailability in the gastrointestinal tract limit their activity and health benefits [102]. Likewise, topical use of polyphenols is limited because of their quick oxidation, which leads to food browning and formation of disagreeable odors that is accompanied by a loss in activity [102]. In relation to their free form, phenolic compounds show a limited solubility in water and have an unpleasant taste, which must be masked before their incorporation into foodstuff or oral medicines [102].

Nowadays, new forms of administration, such as nano encapsulation, prodrugs, and phytosomes, have been developed in an attempt to control and maintain the release of phenolic compounds in the gastrointestinal tract, improve their solubility and increase their bioavailability.

### 4.1. Nano-Encapsulation

One of the most important applications of nanotechnology is nano-encapsulation of bioactive compounds. Nano-encapsulation improves phenolics protection and bioavailability due to increase surface-volume ratio by reducing particle size to the nanoscale [103]. There are different phenolic nanoscale delivery systems such as nano-encapsulation of phenolics with biopolymer-based technologies, nano-encapsulation of phenolics by natural nano-carriers, and nano-encapsulation technologies for phenolics based on special equipment.

#### 4.1.1. Nano-Encapsulation of Phenolics with Biopolymer-Based Technologies

On one hand, polymeric nanoparticles represent a promising phenolic delivery system. They are considered sub-micron particles that can be used for nano-encapsulation of bioactive compounds since they present positive properties like good biocompatibility, easy design and preparation, and biomimetic features [103]. In this sense, it has been demonstrated that nanoparticulation of curcumin into poly(lactic-co-glycolic acid) nanoparticles enhanced curcumin retention time in the cerebral cortex and hippocampus of rats due to the increased oral bioavailability of this phenolic compound [104].

On the other hand, another biopolymer-based delivery system is constituted by nano polymer complexes, which can be manufactured by assembling two different biopolymers such as proteins and polysaccharides. During assembly, both composition of the system and variables affecting the process, such as pH, ionic strength, biopolymer concentration and protein-polysaccharide ratio, must be completely controlled [103]. Previous studies have reported that bovine serum albumin-carrageenan nanoparticles were used as a protective carrier of EGCG, showing a good stability after one month of storage, most likely, because of their extremely small particle size [105]. Moreover, EGCG was also encased into gelatin-based nanoparticles, which kept their biological activity and blocked invasive potential of breast cancer cell line MBA-MD-231 [106].

Finally, polymeric nano-micelles, which can be fabricated by both chemical and physical methods, are used for nano-encapsulation of amphiphilic and poorly water soluble phenolics [88]. For instance, curcumin nanomicelles have been demonstrated to improve cellular caption, in vitro cytotoxicity, and in vitro antiangiogenic effect of curcumin [107].

#### 4.1.2. Nano-Encapsulation of Phenolics by Natural Nano-Carriers

Some examples of widely applied natural nanocarriers are cyclodextrins, casein, chitosan, and nanocrystals. As for cyclodextrins, they are cyclic oligosaccharides constituted by 6, 7 or 8 glucopyranose units joined by a glycosidic bond which improve bioavailability, stability and other functional properties of bioactive compounds [103]. For instance, the inclusion of curcumin in β-cyclodextrin was proven to enhance its delivery and therapeutic efficacy in C4-5 and DU145 prostate cancer cells in comparison to free curcumin [108].

Casein is a physiologically relevant family of phosphoproteins as they concentrate, stabilize, and transport essential nutrients for the neonate. Caseins are very flexible, adapting their conformation to different environmental conditions due to a high amount of proline residues in their structures [103]. These proteins can form micellar nanostructures which could be used as a carrier system for hydrophobic therapeutic agents such as curcumin. In fact, the encapsulation of curcumin in β-casein exhibited higher solubility, cytotoxicity and antioxidant activity than free curcumin in human leukemia cell line K-562 [109].

Regarding chitosan, it is an abundant natural aminopolysaccharide which has degradable, biocompatible, antibacterial, film-forming properties and low toxicity. Chitosan nanoparticles adhere to gastrointestinal tract for a long time due to their mucoadhesive properties, which helps to enhance the bioavailability of bioactive compounds [110]. It has been demonstrated that the encapsulation of EGCG in chitosan-tripolyphosphate nanoparticles improved oral absorption, thus increasing the concentrations of EGCG in the stomach and jejunum of Swiss outbred mice [111]. Besides, atherosclerotic effects of EGCG in rabbits were also enhanced by the encapsulation of EGCG in chitosan nanoparticles [112]. 

Finally, nanocrystals are nanoparticles with a crystalline character. In this case, they are completely composed of phenolic compounds and, therefore, there is not carrier material. Nano-crystallization increases bioavailability of bioactive compounds by improving dissolution velocity because of surface area enhancement and increase in saturation solubility [103]. Thus, rutin nanocrystals have been reported to display a better dissolution behavior, which led to a better bioavailability in the body compared to the poorly soluble free rutin [113]. Furthermore, the antioxidant activity and reducing power of quercetin were dramatically increased when loaded into nanocrystals [114].

#### 4.1.3. Nano-Encapsulation Technologies for Phenolics Based on Special Equipment

Electrospinning is an efficient technique for production of polymer nanofibers by means of electrostatic forces. There are several parameters that can affect this process including solution properties (viscosity, elasticity or molecular weight, among others), processing conditions (hydrostatic pressure, volume feed rate, etc.) and ambient parameters (humidity or solution temperature, inter alia) [103]. For instance, it has been informed that gallic acid was successfully loaded into zein ultra-fine fibers by using electro-spinning, gallic acid retaining its antioxidant activity after its incorporation into zein nanofibers [115].

Electrospraying is a similar process to electrospinning but, in this case, the application of an electric field generates droplets instead of nanofibers [103]. This technique has been used to encapsulate EGCG, among other bioactive compounds. Importantly, EGCG antioxidant activity was fully retained for 100 h after encapsulation [116].

Finally, nano-spray drying is a mechanical method which allows manufacturing and collecting submicron droplets from a solution [103]. In this sense, recent studies have described that curcumin was successfully encapsulated in chitosan nanoparticles using spray drying process and it was showed that curcumin is totally released from particles within a 2 h period [117].

### 4.2. Prodrugs

Prodrugs are inactive compounds created by chemical modification of bioactive compounds. Prodrugs are usually transformed into the active form in single-step hydrolysis of the modified groups [118]. This approach is used to improve the oral bioavailability of drugs which are poorly absorbed through the gastrointestinal tract, as well as to enhance drug targeting [119]. In this line, resveratrol prodrugs have been proved to ameliorate colon inflammation in a murine model of dextran sulfate sodium-induced colitis. Particularly, mice fed with resveratrol prodrugs were reported not to develop colitis due to protection from rapid metabolization and excretion of resveratrol and, subsequently, a more effective delivery of anti-inflammatory resveratrol to the colon [120].

### 4.3. Phytosomes

Phytosomes are lipid-molecule complexes which are obtained by binding a herbal extract, or their constituents, to phospholipids, mainly phosphatidylcholine [121]. Such complexes have demonstrated to successfully enhance bioavailability by providing an environment of improved lipophilicity. For example, a catechin-phospholipid complex exhibited a sustained release over 24 h in an in vitro dissolution study and better antioxidant activity than free catechin at all doses tested [122]. In addition, a naringenin-phospholipid complex showed a high drug content and a better drug release at the end of the in vitro dissolution study in distilled water compared to free naringenin [123].

## 5. Concluding Remarks

Phenolic compounds are secondary metabolites ubiquitously present in the plant kingdom that have become a research hot topic over the last ten years. They can be classified according to different systems but, regardless of the classification used, all phenolics possess an aromatic ring with at least one hydroxyl substituent. The number and position of the hydroxyl groups on the chemical structure determine the potential of phenolic compounds as antioxidant molecules. Their ability to attenuate ROS production and chelate metal ions has been reported in numerous in vitro and in vivo models. Moreover, these compounds exert a wide spectrum of biological effects, from antimicrobial to anticancer activities, as well as protective effects against neurodegenerative diseases such as Alzheimer’s or Parkinson’s. The pleiotropic biological properties of dietary polyphenols are however limited by their bioavailability since there is an important divergence between theoretical polyphenol composition of food, their bioactivity, and circulating levels reached after their consumption. To deal with this challenge, novel delivery systems have emerged in the last few years for enhancing bioavailability of phytochemicals. Most of these methods are based in nanotechnology and are intended to provide protection to phenolics from microenvironmental conditions such as pH, light, oxygen, and temperature. Thus, nano-encapsulation is an encouraging delivery system for the efficient transportation and release of phenolic compounds to target tissues. Similarly, bioavailability issues may be also overcome by manufacturing phenolics as prodrugs or phytosomes. All these promising strategies can be very appealing for the pharma and functional food industries because modified phenolics may serve as bioactive drugs or be incorporated into formulations of functional food products like beverages. Nevertheless, future research should focus on biological fate of encapsulated phenolic compounds after absorption, distribution, metabolism, and excretion in the human body. Furthermore, clinical studies are warranted to determine effective concentration, bioavailability, efficacy, and toxicity/safety profile of encased phenolics for therapeutic purposes.

## Figures and Tables

**Figure 1 antioxidants-09-01263-f001:**
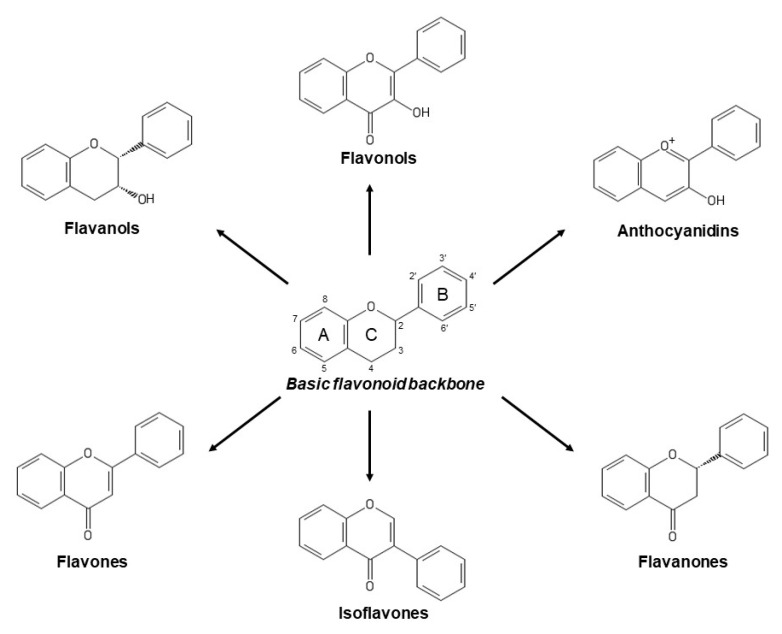
Basic flavonoid structure and main types of flavonoids.

**Figure 2 antioxidants-09-01263-f002:**
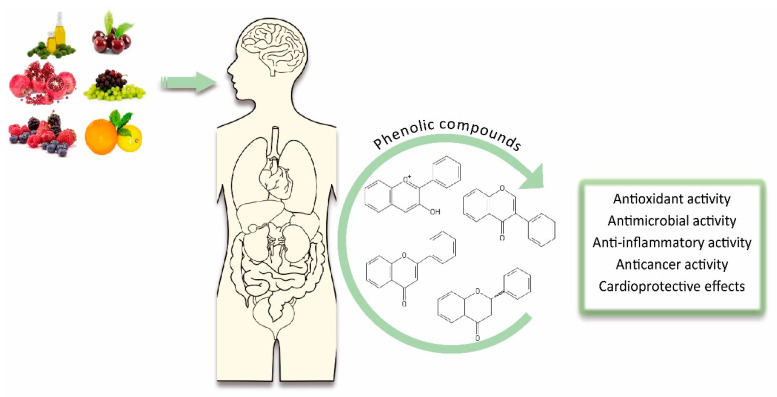
An overview on the biological effects of dietary phenolic compounds.

**Table 1 antioxidants-09-01263-t001:** Main factors affecting bioavailability of phenolic compounds.

Type of Factor		References
Food-related factors	Food processing (thermal treatments, lyophilization, homogenization, cooking, and culinary methods)	[15,17]
	Interactions between food components (fiber, proteins, polysaccharides, fat, bond with proteins)	
Host-related factors	Systemic factors (age, gender, and genetics)	[15]
	Intestinal factors (enzymatic activity and colonic microflora)	
Phenolics-related factors	Chemical structure (molecular weight, lipophilicity, and stereochemistry)	[18]
Other factors	Transport mechanisms (passive diffusion, facilitated diffusion, and active transport)	[15,16]
	External factors (ripening stage, cultivars, and agronomic conditions)	

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
