# Peer review of "Plant Phenolics: Bioavailability as a Key Determinant of Their Potential Health-Promoting Applications"

_antioxidants, 2020, doi:10.3390/antiox9121263_

Round 1

Reviewer 1 Report

This a well-constructed paper summarizing already quite well described polyphenols activities. I am concerned about its scientific importance of this paper is quite low.  

There are so many (including recently published) reviews about polyphenols and their “health-promoting potential” (https://doi.org/10.1016/j.foodchem.2020.127535;  https://doi.org/10.1080/10942912.2017.1354017; https://doi.org/10.3390/nu12082190), that any new one has to face great demands.

First of all, polyphenols are such a broad topic and there are many papers describing their main biological potential and applications. New paper is expected to provide insightful and critical analysis of the subject and the latest reports. Meanwhile, sections ‘1. Introduction’ and ‘3. Biological effect…’ contain quite general information, and are collections of quite commonly known information. What about doses, toxicity, clinical trials, detailed mechanism? Therefore, the section ‘4. Current trends…” is the most interesting part of the manuscript.

The second major issue is overestimating the role of role of polyphenols as direct antioxidants (e.g. L. 15-19; L. 218-229). The issue of indirect antioxidant activity of polyphenols cannot be ignored. https://www.teknoscienze.com/getpdf.php?filename=Contents/Riviste/PDF/AF2_2012_RGB.pdf&beginpage=36&endpage=39&filetitle=New%20antioxidant%20mechanisms%20and%20functional%20foods%20Part%202.

Author Response

This a well-constructed paper summarizing already quite well described polyphenols activities. I am concerned about its scientific importance of this paper is quite low. 

There are so many (including recently published) reviews about polyphenols and their “health-promoting potential” (https://doi.org/10.1016/j.foodchem.2020.127535;  https://doi.org/10.1080/10942912.2017.1354017; https://doi.org/10.3390/nu12082190), that any new one has to face great demands.

- First of all, polyphenols are such a broad topic and there are many papers describing their main biological potential and applications. New paper is expected to provide insightful and critical analysis of the subject and the latest reports. Meanwhile, sections ‘1. Introduction’ and ‘3. Biological effect…’ contain quite general information, and are collections of quite commonly known information. What about doses, toxicity, clinical trials, detailed mechanism? Therefore, the section ‘4. Current trends…” is the most interesting part of the manuscript.

Based on the Reviewer’s comments we have added a new subsection to cover aspects related to doses and toxicity of phenolic compounds (3.6. Safety profile of phenolic compounds; lines 442-463). Likewise, despite the description of some clinical studies were explained in the original version of the manuscript (lines 402-408 and 418-420), new information has been included to further address the effect of phenolic compounds in clinical trials (lines 358-365 and 428-431).

On the other hand, given the variety and heterogeneity of existing phenolic compounds, detailed mechanism is a complex topic that goes far beyond the scope of the present review. In any case, we already outlined some signaling pathways involved in the effects of phenolic compounds (e.g., lines 305-317, 352-355, or 413-418) in the first version of the manuscript.

As for general comments, we agree with the reviewer that sections 1 and 3 contain general information. In fact, that is what an Introduction is supposed to be, i.e., a compilation of already known information on the topic to put the reader in the picture. Moreover, the present manuscript is intended to reach a broad audience, not only catch the attention of specialized readers. That is why we have tried to give a wide, updated overview of the biological effects of phenolic compounds in section 3. Anyway, as we are aware that the most interesting parts of the manuscript are those related to the bioavailability of phenolic compounds, we have made some changes in the title (highlighted in red) of the manuscript as we think that it now better reflects the content of the manuscript and emphasizes the relevance of bioavailability in the biological effects of phenolic compounds.

- The second major issue is overestimating the role of role of polyphenols as direct antioxidants (e.g. L. 15-19; L. 218-229). The issue of indirect antioxidant activity of polyphenols cannot be ignored.

https://www.teknoscienze.com/getpdf.php?filename=Contents/Riviste/PDF/AF2_2012_RGB.pdf&beginpage=36&endpage=39&filetitle=New%20antioxidant%20mechanisms%20and%20functional%20foods%20Part%202.

According to the Reviewer´s suggestion we have rephrased some parts of the text to clarify the importance of indirect antioxidant activity of polyphenols (lines 15-17, 234-237, 245, and 263). Nevertheless, we would like to emphasize that this aspect was already mentioned in the original version of the manuscript (lines 237-238, 239-240, 263-264, and 379-381).

Reviewer 2 Report

This is a comprehensive review on phenolics physiological effects with special emphasis given to the influence of their bioavailability. Paper is well written and summarizes the general knowledge on health benefits of these compounds. Paper could be published as it is, however, I have a feeling that some more attention should be given to synergetic action of these compounds, however, I do know that it is not well studied subject.

Author Response

This is a comprehensive review on phenolics physiological effects with special emphasis given to the influence of their bioavailability. Paper is well written and summarizes the general knowledge on health benefits of these compounds. Paper could be published as it is, however, I have a feeling that some more attention should be given to synergetic action of these compounds, however, I do know that it is not well studied subject.

Following the Reviewer’s suggestion, we have rephrased some parts of the manuscript (lines 250-257 and 342-344) to highlight the importance of synergetic actions of polyphenols. Anyway, we would like to highlight that this aspect was already mentioned in the original version of the manuscript to some extent (lines 279-285, 348-350, and 355-357).

Reviewer 3 Report

The work is current and is adequately presented. However, the introduction is limited, and the presentation of the compound families should be expanded. In particular, non-flavonoid compounds are presented inadequately. I suggest an extension of the paragraph.

Author Response

The work is current and is adequately presented. However, the introduction is limited, and the presentation of the compound families should be expanded. In particular, non-flavonoid compounds are presented inadequately. I suggest an extension of the paragraph.

According to the Reviewer suggestion, we have extended the classification of non-flavonoids (lines 79-105).

Round 2

Reviewer 1 Report

The manuscript has benefited from the revision process and, in general, the issues pointed out by the referees have been adequately addressed.